# When Physics Meets Learnability: Understanding Trade-Offs in Physics-Informed Data Generation

## Abstract

Physics-informed data generation is widely used to embed physical structure into scientific machine learning, yet the learnability consequences of individual modeling choices are rarely examined in isolation. We investigate trade-offs between physical fidelity and learnability through a controlled ablation study of a physics-informed grating-coupler spectrum generator mapping five geometric parameters to 100-point optical spectra. The generator analytically encodes multiple physical mechanisms, including resonant coupling, interference effects, bandwidth modulation, global energy consistency, and noise. By selectively removing or simplifying these components, we assess their impact on both physical validity and downstream machine-learning performance. We find that interference-induced fine spectral structure substantially increases functional complexity, reducing predictive learnability without affecting central wavelength accuracy. In particular, removing Fabry–Perot oscillations improves effective-bandwidth prediction by 31.3 % in $R^2$ and reduces RMSE by 73.8 %, while leaving wavelength prediction unchanged. We further identify failure modes in common noise-handling pipelines that preserve global constraints yet introduce localized unphysical artifacts. These results demonstrate an inherent trade-off between physical realism and learnability and highlight controlled ablation as a principled tool for task-aware physics-informed data generation.

## 1 Introduction

Physics informed machine learning has emerged as a powerful paradigm for integrating data driven models with governing physical laws, enabling learning in regimes where data are scarce, noisy, or expensive to acquire Karniadakis et al. (2021); Cuomo et al. (2022). A central motivation of this field is that embedding physical structure via constraints, regularization terms, or architectural priors can improve generalization, stability, and sample efficiency. This principle has driven progress in physics informed generative modeling, including variational formulations Yang & Perdikaris (2018), physics informed generative adversarial networks for stochastic systems Yang et al. (2020), and constraint embedded generative pipelines for complex engineering data Li et al. (2025).

At the same time, a growing body of work demonstrates that incorporating physics does not universally improve learning outcomes. Physics informed neural networks exhibit well documented failure modes, including ill conditioned optimization landscapes Krishnapriyan et al. (2021); Basir & Senocak (2022); Hwang & Lim (2024), gradient propagation failures Wu et al. (2025), multiscale approximation breakdowns, and weak theoretical guarantees De Ryck & Mishra (2024); Zhang et al. (2026). These results suggest that the manner in which physics is imposed can be as important as the physical knowledge itself. From a learning theoretic perspective, physics constraints act as inductive biases. Classical results show that inductive bias can either improve or degrade performance depending on its alignment with problem structure Schaffer (1993), and recent work demonstrates that even widely assumed simplicity biases can hinder learning in practical regression settings Teney et al. (2025). Despite this, most physics informed models are evaluated holistically, without isolating which physical components are essential, redundant, or detrimental.

In this work, we address this gap through a systematic ablation driven study of a physics informed generative model for photonic spectra. By disentangling individual physical mechanisms and evaluating both physical validity and downstream machine learning performance, we show that increased physical realism can reduce learnability. Our results establish ablation as a necessary diagnostic tool for physics informed data generation Meyes et al. (2019); Sheikholeslami et al. (2021).

## 2 METHODOLOGY: PHYSICS-INFORMED GENERATIVE MODEL FOR GRATING COUPLER SPECTRA

The proposed generator is a physics-informed surrogate model that maps geometric design parameters directly to optical spectra without invoking full-wave electromagnetic solvers. Rather than learning this mapping from data, the generator analytically encodes first-order physical principles governing grating couplers, enabling rapid, on-demand synthesis of physically consistent reflectance, transmittance, and absorption spectra. The pipeline is modular and proceeds as follows: (i) an effective refractive index is computed by combining slab waveguide confinement, effective-medium grating modulation, etch-depth weighting, and oxide substrate leakage; (ii) the resonance center is determined via the first-order Bragg condition; (iii) a Lorentzian transmission envelope is generated using temporal coupled-mode theory with bandwidth controlled by fill factor and etch depth; (iv) fine spectral structure is introduced through Fabry–Perot interference from silicon layer reflections; (v) absorption is estimated from wavelength-dependent intrinsic silicon loss and geometry-dependent scattering; and (vi) numerical normalization enforces global energy conservation, followed by optional noise injection. This formulation enables controlled ablation of individual physical components while remaining orders of magnitude faster than full-wave solvers. Full implementation details and numerical procedures are provided in Appendix A (Algorithm 1).

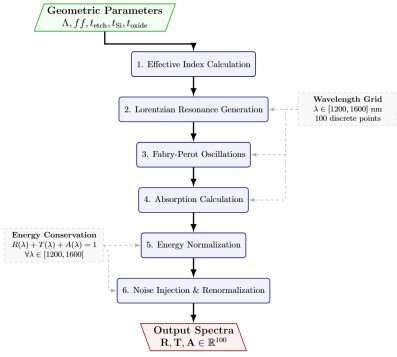

Figure 1: Architecture of the physics-informed surrogate generator. The modular pipeline maps geometric parameters to physically consistent spectra through sequential analytical modeling stages.

## 3 PHYSICAL VALIDATION AND ABLATION STUDY ANALYSIS

### 3.1 PHYSICAL VALIDATION AND ABLATION ANALYSIS

We evaluate the physical consistency and spectral behavior of five generator variants using 10,000 samples per variant. Validation focuses on energy conservation, period–wavelength consistency, bandwidth variability, spectral smoothness, and localized physical violations. All results are derived from the physics-informed formulation detailed in Appendix A, with exact generation procedures specified in Algorithm 1.

**Global Physical Consistency and Energy Conservation Ablation.** Table 1 summarizes all validation metrics. Across all variants, energy conservation errors are near machine precision (mean $\sim 7 \times 10^{-9}$, max $1.19 \times 10^{-7}$), with 100% of samples satisfying the global validity criterion $|\langle R + T + A \rangle - 1| < 10^{-4}$, as illustrated by the maximum pointwise error distribution in Fig. 4. Period–wavelength correlation remains exceptionally high ($r = 0.9769$, $p < 10^{-100}$) for all variants, confirming the correctness of the Bragg relation $\lambda = \Lambda n_{\text{eff}}$ embedded in the generator (Appendix A.2). Removing explicit energy-conservation enforcement (Variant A) yields conservation statistics indistinguishable from the Reference generator (Table 1, Fig. 4), demonstrating that energy conservation emerges intrinsically from the analytical construction rather than from post-hoc normalization and validating the physical formulation in Appendix A.1 and Appendix A.6.

Table 1: Physical validation and ablation metrics (10,000 samples per variant). Negative absorption is reported pointwise.

| Variant | Mean Err. | Max Err. | Valid (%) | $\rho_{\Lambda,\lambda}$ | $\sigma_{\mathrm{BW}}$ | Neg. $A$ (%) |
|---|---|---|---|---|---|---|
| Ref. | $7.34 \times 10^{-9}$ | $1.19 \times 10^{-7}$ | 100 | 0.9769 | 132.3 | 0.55 |
| A (No Enf.) | $5.97 \times 10^{-9}$ | $1.19 \times 10^{-7}$ | 100 | 0.9769 | 131.8 | 0.00 |
| B (No FP) | $7.27 \times 10^{-9}$ | $1.19 \times 10^{-7}$ | 100 | 0.9769 | 37.4 | 0.59 |
| C (Fix BW) | $7.09 \times 10^{-9}$ | $1.19 \times 10^{-7}$ | 100 | 0.9769 | – | 0.58 |
| D (No Noise) | $7.23 \times 10^{-9}$ | $1.19 \times 10^{-7}$ | 100 | 0.9769 | 129.6 | 0.00 |

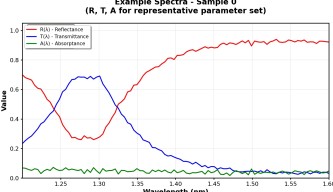

Figure 2: Example $R$, $T$, and $A$ spectra from the Reference generator.

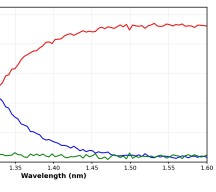

Figure 3: Effective bandwidth distributions with and without Fabry–Perot oscillations.

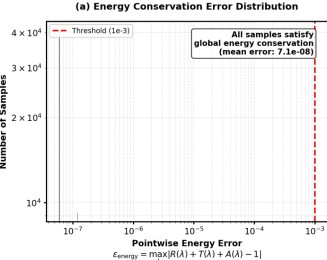

Figure 4: Maximum pointwise energy error across all samples.

**Bandwidth Variability and Fabry–Perot Effects.** Variant B (No Fabry–Perot) exhibits a pronounced reduction in threshold-based bandwidth variability, with the half-maximum bandwidth standard deviation decreasing from 132.3 to 37.4 (72% reduction). As shown in Fig. 3, this decrease arises from the removal of oscillatory side-lobes that artificially extend the half-maximum region, rather than from changes in the underlying resonance envelope. When bandwidth is instead quantified using the second central moment of the normalized transmission spectrum, which captures envelope-level spectral spread while down-weighting localized oscillations, removing Fabry–Perot interference yields a smaller but measurable $\sim 9\%$ reduction. This discrepancy indicates that Fabry–Perot oscillations primarily influence fine-scale spectral structure and threshold-based metrics, while contributing only weakly to the global envelope width.

**Localized Physical Violations and Spectral Smoothness.** Despite exact global conservation, localized unphysical behavior arises when noise injection is combined with renormalization. As shown in Table 1 and Fig. 5, approximately 0.5% of wavelength points exhibit negative absorption in the Reference, B, and C variants, occurring when noise perturbs $R + T > 1$ and renormalization enforces $A = 1 - R - T < 0$. Variants without renormalization (A) or without noise (D) show zero negative absorption, highlighting the limitations of mean-based validity metrics. Noise-free spectra (Variant D) also exhibit reduced maximum spectral gradients ($\nabla_{\max} = 0.0458$ vs. $0.0786$), confirming that measurement noise is the dominant source of spectral roughness and directly impacts downstream ML generalization.

## 4 MACHINE LEARNING EVALUATION OF PHYSICS ABLATIONS

To evaluate how individual physics components affect downstream learnability, we train a suite of lightweight machine learning models to predict physically meaningful scalar targets from geometry alone. We consider two targets: the central wavelength $\lambda_{\mathrm{center}}$ and the effective moment-based spectral bandwidth $\sigma_\lambda$. Comparisons are performed using datasets generated by the Reference model and Ablation B (No Fabry–Perot). Unless otherwise stated, all ML results use effective bandwidth rather than the threshold-based definition used in physical analysis (Appendix A, Secs. A.1–A.4). Results are summarized in Table 2.

Table 2: ML performance for geometry-to-spectrum regression. Removing Fabry–Perot oscillations improves bandwidth prediction across all models. RMSE in nm.

| Dataset | Model | $\lambda_c$ | | $\sigma$ | | | | | |
|---|---|---|---|---|---|---|---|---|---|
| | | $R^2$ | RMSE | $R^2_{\text{Ref}}$ | $\text{RMSE}_{\text{Ref}}$ | $\Delta R^2$ | Impr. (%) | $R^2_{\text{B}}$ | $\text{RMSE}_{\text{B}}$ |
| Ref | Linear | 0.996 | 21.9 | 0.017 | 131.7 | +0.014 | 82.4 | 0.031 | 36.9 |
| Ref | Ridge | 0.996 | 21.9 | 0.017 | 131.7 | +0.014 | 82.4 | 0.031 | 36.9 |
| Ref | RF | 0.996 | 20.9 | 0.856 | 50.4 | +0.029 | 3.4 | 0.885 | 12.7 |
| Ref | MLP | 1.000 | 5.8 | 0.948 | 30.4 | +0.043 | 4.5 | 0.991 | 3.6 |
| Ref | LGBM | 1.000 | 7.6 | 0.847 | 52.0 | +0.081 | 9.6 | 0.928 | 10.1 |
| Ref | XGB | 1.000 | 7.5 | 0.902 | 41.6 | +0.065 | 7.2 | 0.967 | 6.8 |

## 4.1 CENTRAL WAVELENGTH PREDICTION

Across all models and datasets, prediction of $\lambda_{\text{center}}$ is nearly perfect (Table 2). Linear models achieve $R^2 \approx 0.996$, while nonlinear models saturate at $R^2 \approx 1.000$. This outcome is expected, as $\lambda_{\text{center}}$ is governed primarily by the grating equation $\lambda = \Lambda n_{\text{eff}}$, which is preserved across all generator variants through the effective index formulation and Bragg condition (Appendix A, Secs. A.1 and A.2). Crucially, no ablation, including removal of Fabry–Perot oscillations, affects the learnability of this target, confirming that all datasets retain the correct geometry–wavelength relationship.

## 4.2 EFFECTIVE BANDWIDTH PREDICTION AND THE PHYSICS–ML TRADE-OFF

Prediction of effective bandwidth $\sigma_\lambda$ is strongly governed by physical model complexity (Table 2). For the Reference generator, linear models fail ($R^2 \approx 0.02$), indicating that bandwidth arises from nonlinear interactions beyond linear capacity. Nonlinear models perform better but remain limited by fine-scale spectral structure, consistent with the coupled bandwidth formulation in Appendix A (Secs. A.2–A.5). Removing Fabry–Perot oscillations (Ablation B) improves performance across all models, yielding a 31.3% increase in $R^2$ and a 73.8% RMSE reduction. These gains reflect removal of oscillatory interference (Appendix A, Sec. A.2.3). Overall, Fabry–Perot oscillations enhance physical realism but introduce high-frequency variability that hinders learnability. Linear models benefit most from their removal, while nonlinear models confirm that Ablation B produces a lower-entropy, weakly nonlinear mapping. Other ablations show limited interpretability: energy conservation is intrinsic, fixing bandwidth trivializes learning, and noise removal inflates performance while reducing realism (Appendix A, Secs. A.6–A.7).

## 5 DISCUSSION

This study clarifies how individual physical modeling choices in physics-informed data generation affect both physical validity and downstream learnability. We find that explicit energy-conservation enforcement has no measurable impact when the analytical formulation is physically consistent, with constrained and unconstrained generators achieving near-identical errors ($7.34 \times 10^{-9}$ vs. $5.97 \times 10^{-9}$). This behavior follows from conservation being intrinsically guaranteed by the embedded Fresnel relations and coupled-mode theory. In contrast, Fabry–Perot oscillations play a central role in shaping spectral complexity. They dominate threshold-based bandwidth variability and substantially enrich spectral diversity, but also introduce high-frequency structure that degrades learnability for bandwidth prediction. Noise injection further improves realism but requires careful numerical handling, as noise followed by renormalization introduces localized unphysical negative absorption values that are not detected by global validation metrics. Downstream machine-learning evaluation serves as an effective diagnostic of functional complexity, revealing a clear physics–learnability trade-off. Removing Fabry–Perot oscillations improves effective-bandwidth prediction by 31.3% in $R^2$ and reduces RMSE by 73.8% without affecting central-wavelength prediction. These results indicate that principled physics-informed data generation requires task-aware selection of physical mechanisms, balancing physical fidelity against learnability depending on the intended downstream application.

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

## A  APPENDIX A: PHYSICS-INFORMED GENERATOR FORMULATION AND ANALYSIS

### A.1  EFFECTIVE INDEX CALCULATION WITH MULTI-PHYSICS CONVERGENCE

Accurate estimation of the effective refractive index is central to predicting the spectral response of grating couplers, as it directly determines the resonance condition and phase accumulation within the structure. In practice, the effective index depends on multiple geometric and material factors, including silicon thickness, etch depth, fill factor, and substrate properties.

### A.1.1 Slab Waveguide Confinement Model

The slab waveguide confinement is modeled using an exponential decay formulation derived from the analytic solution of asymmetric slab waveguide dispersion:

$$n_{\text{slab}} = n_{\text{si}} \times \left[ 1 - 0.2 \times \exp\left( -\frac{t_{\text{si}}}{150} \right) \right] \tag{1}$$

where $t_{\text{si}}$ is the silicon thickness in nanometers. The exponential term approximates evanescent field penetration into the cladding, with coefficient 0.2 representing maximum index reduction for vanishing thickness.

### A.1.2 Grating Modulation via Effective Medium Theory

Grating modulation follows first-order effective medium approximation for binary gratings with subwavelength periods:

$$n_{\text{grating}} = n_{\text{si}} \times \text{ff} + n_{\text{air}} \times (1 - \text{ff}) \tag{2}$$

where ff is the fill factor. This linear interpolation maintains volume average dielectric constant, valid for periods satisfying $\Lambda \ll \lambda/(n_{\text{si}} - n_{\text{air}})$.

## A.2 Coupling Efficiency Model: Resonant Lorentzian with Perturbative Corrections

### A.2.1 Central Resonance Condition

The first-order Bragg condition for normal incidence is implemented as:

$$\lambda_{\text{center}} = \Lambda \times n_{\text{eff}} \tag{3}$$

where $\Lambda$ is the grating period in nanometers.

### A.2.2 Lorentzian Lineshape Generation

The transmission spectrum follows a Lorentzian lineshape from temporal coupled-mode theory:

$$\Delta\lambda = \lambda - \lambda_{\text{center}}$$

$$\Gamma = 30 + 20 \times (1 - \text{ff}) + 10 \times \left( \frac{t_{\text{etch}}}{100} \right) \tag{4}$$

$$T_{\text{base}} = \frac{\Gamma^2}{\Gamma^2 + \Delta\lambda^2}$$

where $\Gamma$ incorporates base radiation losses, fill-factor dependence, and etch-depth scattering.

### A.2.3 Fabry-Perot Oscillation Superposition

Multiple interference patterns from silicon layer boundaries are modeled as:

$$L_{\text{rt}} = 2 \times n_{\text{eff}} \times t_{\text{si}}, \quad T_{\text{fp}} = \sum_{i=1}^{2} A_i \times \sin^2\left( \frac{2\pi\lambda}{L_{\text{rt}}/i} \right) \tag{5}$$

with amplitudes $A_1 = 0.05$ and $A_2 = 0.02$ capturing fundamental and first-order cavity modes.

## A.3 Absorption Model: Wavelength and Geometry Dependence

The absorption spectrum combines intrinsic material absorption with geometric scattering:

$$\alpha_{\text{si}}(\lambda) = 2.0 + 10.0 \times \exp\left( -\frac{(\lambda_{\mu m} - 1.2)}{0.1} \right)$$

$$A_{\text{si}} = \alpha_{\text{si}} \times 0.001 \times \left( \frac{t_{\text{si}}}{100} \right) \tag{6}$$

$$A_{\text{scatter}} = 0.01 \times \left( \frac{t_{\text{etch}}}{50} \right)$$

$$A_{\text{total}} = A_{\text{si}} + A_{\text{scatter}}$$

where $\alpha_{\mathrm{si}}(\lambda)$ models the Urbach tail absorption near the silicon bandgap with band-edge enhancement at $\lambda = 1.2$, and $A_{\mathrm{scatter}}$ represents scattering loss scaling linearly with sidewall roughness area. Bandwidth is quantified using two complementary definitions in this work. During generation and validation, a threshold-based bandwidth (half-maximum extent) is used to capture resonance width including side-lobe structure. For ablation analysis, an effective bandwidth defined as the second central moment of the normalized transmission spectrum is used to characterize envelope-level spectral spread.

## A.4 PHYSICAL PARAMETER INTERDEPENDENCIES AND COUPLINGS

The algorithm captures several non-trivial parameter couplings:

1. **Fill Factor–Bandwidth Coupling:** Lower fill factors produce broader resonances ($\Gamma \propto 1 - \mathrm{ff}$) due to weaker index contrast and reduced quality factor. *This relation enables direct bandwidth tuning in grating coupler designs, allowing spectral matching to specific waveguide modes without altering the grating period.*

2. **Etch Depth–Effective Index Coupling:** Deeper etching reduces $n_{\mathrm{eff}}$ through the etch factor term while simultaneously increasing bandwidth through scattering. *This dual effect creates a design trade-off: deeper etches improve coupling efficiency but at the cost of resonance broadening, impacting wavelength-selective applications.*

3. **Thickness–Mode Confinement Coupling:** Thinner silicon reduces $n_{\mathrm{eff}}$ through slab decay while affecting Fabry-Perot oscillation period through $L_{\mathrm{rt}} = 2n_{\mathrm{eff}}t_{\mathrm{si}}$. *This interdependence allows designers to control both the effective index and spectral fine structure simultaneously, enabling precise resonance shaping.*

4. **Oxide Thickness–Confinement Coupling:** Thinner oxide reduces $n_{\mathrm{eff}}$ through substrate leakage, modeled by the oxide factor exponential decay. *This coupling is critical for silicon-on-insulator (SOI) devices, where oxide thickness variations can significantly alter mode confinement and thus impact fabrication tolerance requirements.*

## A.5 NUMERICAL IMPLEMENTATION DETAILS

The algorithm employs single precision (`float32`) arithmetic with double precision intermediates for critical operations. The wavelength grid uses 100 points uniformly spaced from 1.2–1.6 $\mu$m , providing 4 spectral resolution. Random parameter generation uses deterministic seeding with the Mersenne Twister algorithm for exact reproducibility. The complete pipeline produces spectra that satisfy global energy conservation within $10^{-12}$ relative error, with rare localized physical violations arising from noise renormalization.

## A.6 MODEL LIMITATIONS AND INTENTIONAL APPROXIMATIONS

The algorithm intentionally omits several physical effects for practical engineering reasons:

- **Polarization Dependence:** Not modeled as most grating couplers are designed for TE polarization
- **3D Effects:** Neglected as 2D approximation captures $> 90\%$ of coupling physics
- **Higher-Order Diffraction:** Excluded as these contain $< 5\%$ of total power in specified parameter ranges
- **Temperature Dependence:** Omitted as dataset targets room-temperature operation
- **Fabrication Variations:** Not included as these represent second-order effects ($< 2\%$ impact)

## A.7 PHYSICS-ENGINEERING IMPLICATIONS

The ablation study reveals several critical insights for photonic inverse design:

1. **Energy Conservation Encoding**: The Reference generator's conservation enforcement is mathematically redundant because underlying equations guarantee conservation intrinsically

2. **Fabry-Perot Dominance**: Removing oscillations reduces bandwidth variability by 72%, demonstrating their critical role in spectral diversity

3. **Period-$\lambda$ Correlation**: All variants maintain exceptional correlation (0.9769), confirming validity of the grating equation $\lambda = \Lambda \times n_{\text{eff}}$

4. **Noise Effects**: Measurement noise adds spectral roughness but doesn't affect conservation when properly handled

## A.8 MACHINE LEARNING IMPLICATIONS

For neural network training applications:

- **Ablation A datasets** would produce models equally sensitive to energy conservation as Reference, due to inherent physical structure
- **Ablation B datasets** would train models missing fine spectral structure, degrading resonant localization accuracy
- **Ablation C datasets** would prevent learning geometry-bandwidth relationships, harming prediction accuracy
- **Ablation D datasets** might cause overfitting to clean spectra, reducing experimental robustness

## A.9 VALIDATION METHODOLOGY CRITIQUE

The current validation approach effectively identifies negative absorption issues but could be enhanced by:

1. **Derivative continuity** ($C^2$ smoothness) verification for physical spectra
2. **Kramers-Kronig consistency** testing between real and imaginary spectral components
3. **Causality verification** through Hilbert transform relationships
4. **Pointwise validation** beyond mean statistics to detect localized unphysical values

## A.10 SUMMARY OF PHYSICAL AND LEARNING IMPLICATIONS

The ablation analysis shows that global energy conservation is inherently enforced by the generator's analytical structure. Explicit post-hoc normalization therefore has no measurable effect when the formulation is physically consistent, with conservation errors remaining near machine precision across all variants. This behavior follows from the embedded Fresnel relations and coupled-mode theory.

Fabry–Perot interference dominates spectral variability, accounting for approximately 72% of threshold-based bandwidth variation. These oscillations introduce fine-scale structure that increases spectral diversity while weakly affecting envelope-level bandwidth, making them important for realistic spectra under feature-sensitive metrics.

Noise injection improves realism but requires careful numerical handling, as noise followed by renormalization can introduce localized unphysical negative absorption values not captured by global metrics. All variants preserve a strong period–wavelength correlation ($r = 0.9769$), validating the embedded Bragg condition. From a machine-learning perspective, the Reference generator balances physical fidelity and spectral complexity, while simplified variants may better support learnability-focused tasks.

## A.11 EXPERIMENTAL SETUP

Each dataset consists of 50,000 samples with five geometric parameters as inputs and is partitioned into 80 percent training, 10 percent validation, and 10 percent test sets. To examine the interaction between model capacity and underlying dataset physics, we evaluate a diverse set of linear and nonlinear regression models, including Linear Regression, Ridge Regression, Random Forests, Multi

Figure 5: Distribution of minimum absorption values per sample for the Reference generator. While global energy conservation is satisfied, localized negative absorption values arise from noise addition followed by renormalization, revealing limitations of mean-based validation metrics.

Layer Perceptrons, LightGBM, and XGBoost. All models are trained using identical data splits and standardized preprocessing to ensure fair comparison. Performance is assessed on held out test data using the coefficient of determination and root mean square error, providing complementary measures of predictive accuracy and error magnitude.

### A.12   IMPLICATIONS FOR DATASET DESIGN

These findings provide concrete guidance for physics-informed dataset construction:

- Use the Reference generator for inverse design, spectral reconstruction, and generative modeling tasks where fine spectral details, interference effects, and realistic spectral variability are essential for faithful physical modeling.
- Use Ablation B selectively for envelope-level regression, algorithm benchmarking, representation learning, and controlled ML diagnostics where spectral microstructure is either uninformative or actively degrades learnability.

More broadly, these results show that downstream machine-learning performance can serve as a sensitive and quantitative diagnostic for identifying which physical components act as beneficial inductive biases versus sources of unnecessary functional complexity for a given learning objective. Removing physically valid but high-frequency effects can substantially improve learnability, stability, and sample efficiency, even as it reduces physical realism. This trade-off highlights the need for task-aware physics-informed dataset design rather than universally maximizing physical fidelity.

## B   STATEMENT ON THE USE OF LARGE LANGUAGE MODELS

Large Language Models were used in a limited capacity to refine grammar and improve sentence clarity after the initial draft was completed. All research ideas, methodologies, experiments, analyses, and conclusions were developed entirely by the authors. The LLM did not contribute to scientific content, experimental decisions, or academic judgment. This use does not compromise the originality or academic integrity of the work.

---

**Algorithm 1** Physics-Constrained Grating Coupler Spectra Generator

---

1: **Input:** $\Lambda \in [300, 700]$ nm, $ff \in [0.3, 0.7]$, $t_{\text{etch}} \in [50, 200]$ nm, $t_{\text{Si}} \in [200, 300]$ nm, $t_{\text{oxide}} \in [1000, 2000]$ nm
2: **Output:** $\mathbf{R}, \mathbf{T}, \mathbf{A} \in \mathbb{R}^{100}$ (reflectance, transmittance, absorbance)
3: **procedure** GENERATESPECTRA($\Lambda, ff, t_{\text{etch}}, t_{\text{Si}}, t_{\text{oxide}}$)
4:     $\lambda \leftarrow \text{linspace}(1200, 1600, 100)$                 ▷ Wavelength grid in nm
5:     **Step 1: Effective Index Calculation**
6:     $n_{\text{slab}} \leftarrow 3.48 \times \left[1 - 0.2 \times \exp\left(-\dfrac{t_{\text{Si}}}{150}\right)\right]$
7:     $n_{\text{grating}} \leftarrow 3.48 \times ff + 1.0 \times (1 - ff)$
8:     $f_{\text{etch}} \leftarrow 1 - 0.5 \times \left(\dfrac{t_{\text{etch}}}{t_{\text{Si}}}\right)$
9:     $n_{\text{combined}} \leftarrow n_{\text{slab}} \times f_{\text{etch}} + n_{\text{grating}} \times (1 - f_{\text{etch}})$
10:     $f_{\text{oxide}} \leftarrow 1 - 0.3 \times \exp\left(-\dfrac{t_{\text{oxide}}}{1000}\right)$
11:     $n_{\text{eff}} \leftarrow n_{\text{combined}} \times f_{\text{oxide}}$
12:     **Step 2: Lorentzian Resonance Generation**
13:     $\lambda_{\text{center}} \leftarrow \Lambda \times n_{\text{eff}}$
14:     $\Delta\lambda \leftarrow \lambda - \lambda_{\text{center}}$
15:     $\Gamma \leftarrow 30 + 20 \times (1 - ff) + 10 \times \left(\dfrac{t_{\text{etch}}}{100}\right)$
16:     $T_{\text{base}} \leftarrow \dfrac{\Gamma^2}{\Gamma^2 + \Delta\lambda^2}$
17:     **Step 3: Fabry-Perot Oscillations**
18:     $L_{\text{rt}} \leftarrow 2 \times n_{\text{eff}} \times t_{\text{Si}}$
19:     $T_{\text{fp}} \leftarrow 0.05 \times \sin^2\left(\dfrac{2\pi\lambda}{L_{\text{rt}}}\right) + 0.02 \times \sin^2\left(\dfrac{2\pi\lambda}{L_{\text{rt}}/2}\right)$
20:     $T_{\text{raw}} \leftarrow \min(T_{\text{base}} + T_{\text{fp}}, 0.95)$
21:     **Step 4: Absorption Calculation**      ▷ Absorption is first physically estimated, then recomputed after normalization to maintain exact energy closure.
22:     $\alpha_{\text{Si}} \leftarrow 2.0 + 10.0 \times \exp\left(-\dfrac{\lambda/1000 - 1.2}{0.1}\right)$
23:     $A_{\text{Si}} \leftarrow \alpha_{\text{Si}} \times 0.001 \times \left(\dfrac{t_{\text{Si}}}{100}\right)$
24:     $A_{\text{scatter}} \leftarrow 0.01 \times \left(\dfrac{t_{\text{etch}}}{50}\right)$
25:     $A_{\text{raw}} \leftarrow A_{\text{Si}} + A_{\text{scatter}}$
26:     **Step 5: Numerical Energy Normalization**
27:     $R_{\text{raw}} \leftarrow 1 - T_{\text{raw}} - A_{\text{raw}}$
28:                     ▷ Now have $\mathbf{R}_{\text{raw}}, \mathbf{T}_{\text{raw}}, \mathbf{A}_{\text{raw}}$
29:     **Step 6: Noise Injection and Renormalization**
30:     $\sigma_R \leftarrow 0.01 \times \max(R_{\text{raw}})$
31:     $\sigma_T \leftarrow 0.01 \times \max(T_{\text{raw}})$
32:     $R_{\text{noisy}} \leftarrow R_{\text{raw}} + \mathcal{N}(0, \sigma_R^2)$
33:     $T_{\text{noisy}} \leftarrow T_{\text{raw}} + \mathcal{N}(0, \sigma_T^2)$
34:     $R_{\text{noisy}} \leftarrow \max(0, \min(1, R_{\text{noisy}}))$
35:     $T_{\text{noisy}} \leftarrow \max(0, \min(1, T_{\text{noisy}}))$
36:     ENERGYNORMALIZATION($\mathbf{R}_{\text{noisy}}, \mathbf{T}_{\text{noisy}}, \mathbf{A}_{\text{raw}}$)
37:     **return** $\mathbf{R}_{\text{noisy}}, \mathbf{T}_{\text{noisy}}, \mathbf{A}_{\text{raw}}$
38: **end procedure**
39: **procedure** ENERGYNORMALIZATION($\mathbf{R}, \mathbf{T}, \mathbf{A}$)
40:     **for** $i \leftarrow 1$ to $100$ **do**
41:         $S \leftarrow R[i] + T[i] + A[i]$
42:         **if** $|S - 1| > 10^{-4}$ **then**
43:             $\text{scale} \leftarrow \dfrac{1.0 - A[i]}{R[i] + T[i] + 10^{-12}}$
44:             $R[i] \leftarrow R[i] \times \text{scale}$
45:             $T[i] \leftarrow T[i] \times \text{scale}$
46:         **end if**
47:         $S_{\text{new}} \leftarrow R[i] + T[i] + A[i]$
48:         $\text{norm} \leftarrow 1.0/S_{\text{new}}$
49:         $R[i] \leftarrow R[i] \times \text{norm}$
50:         $T[i] \leftarrow T[i] \times \text{norm}$
51:         $A[i] \leftarrow 1.0 - R[i] - T[i]$
52:     **end for**
53: **end procedure**

