# OpenReview forum: "When Physics Meets Learnability: Understanding Trade-Offs in Physics-Informed Data Generation"
_ICLR.cc/2026/Workshop/Sci4DL — Submitted to Sci4DL 2026_

### Official Review · Reviewer_e9ah · 2026-02-25

**Fit:** 3
**Significance:** 2
**Confidence:** 3

**Summary:**

Paper discusses how     increasing the complex details can actually derail the PIML and     PIDG models. They form a fast     mathematical simulator (grating coupler) which takes in 5 physical measurements and outputs a graph of the light spectrum. Ablation tests are carried out to study the effects of simpler setups (low information) on model performance. They removed the fabry perot oscillation data from their     setup and found the model accuracies improved by around 31% and errors dropped by 73%.

**Strengths:**

1. a Physics informed surrogate model that maps geometric design parameters directly to optical spectra without invoking full electromagnetic solvers.
2. The analytical pipeline and physics ablation studies are very well executed.
3. It is inferred that having high frequency or complex data for PIML/PIDG can actually mislead models  and have inferior performance. A true balance between complexity and  learnability (what a model can actually learn) is required for training.

**Suggestions:**

1. suggested to include more output parameters in their model setup. Predicting just the central wavelength and bandwidth is a very simple regression task.
   2. The  authors fail to include complex light interactions and effects  such as polarization and higher order diffraction in the generator which might   suggest that the trade-off observed may be due to a simpler model.
   3. Only basic ML models  are evaluated. For publication in top-tier AI/ML venues, this is not  sufficient. Authors should have included deep learning models in their  study. In the present setting, it might be just the fact that ML models      don't have sufficient depth in understanding the complex data patterns.

---

### Official Review · Reviewer_yzGc · 2026-02-25

**Fit:** 1
**Significance:** 2
**Confidence:** 2

**Summary:**

This paper conducts a controlled ablation study on a physics-informed surrogate generator for grating-coupler spectra to understand the trade-off between physical realism and machine learning learnability. The authors demonstrate that while certain physical mechanisms, such as Fabry-Perot oscillations, increase physical fidelity, they simultaneously introduce high-frequency functional complexity that degrades the predictive performance of downstream models.

**Strengths:**

1. The overall motivation is clear, starting with questioning the assumption that increasing physical constraints always improves learning outcomes.
2. Their experimental results on the grating-coupler spectra further validate their assumption

**Suggestions:**

1. The investigated dataset is pretty limited, only on the task of grating-coupler spectra. The authors should validate these trade-offs across diverse physical domains.
2. Many of the evaluated metrics show "near-perfect" results, with R^2 ~1.00 and low RMSE for central wavelength prediction. It does not provide evidence on whether the proposed ablations would provide meaningful benefits in more complex, high-dimensional situations.
3. The investigated ML methods seem to be simple. How about more modern methods?
4. Out of scope of this workshop: The paper primarily focuses on "physics-informed data generation" and dataset engineering rather than exploring scientific methods for "deep learning understanding."
5. In Figure 2, the simulated spectrum has many small wrinkles; shouldn't it be smooth? The small up and down may lead to extra uncertainty for your ground truth. Please check the simulation methods.

---

### Official Review · Reviewer_PmnF · 2026-02-25

**Fit:** 1
**Significance:** 1
**Confidence:** 3

**Summary:**

The paper investigates whether adding more physical realism to physics-informed data generation always improves downstream machine learning performance. The authors find that there is a measurable trade-off between physical fidelity and what is called learnability.

**Strengths:**

The paper presents a carefully controlled ablation study on a physically meaningful system (grating couplers). The modular decomposition of physical mechanisms, including Fabry–Perot oscillations, noise injection, and energy normalization, enables a clean isolation of their individual effects. This design is methodologically strong and allows the authors to evaluate how specific physical components influence learnability. In particular, the quantitative demonstration that increasing physical realism can degrade downstream machine learning performance could be a noteworthy and conceptually important contribution.

**Suggestions:**

The paper does not provide an empirical analyses of deep networks, which is the scope of the workshop. It rather focuses on training of shallow neural networks and on their accuracy without proposing insight on the neural networks themselves.

Furthermore, the writing, particularly regarding machine learning concepts, would benefit from greater precision. For example, the notion of “learnability” is not clearly defined. Clarifying this would strengthen the paper. See below additional writing issues.

- l. 029: "highlight controlled ablation as a principled tool for task-aware physics-informed data generation", what is "physics-informed data generation"? As such the sentence is too generic, "controlled ablation" seems to be a principled tool for everything.
- l. 045: what are "learning outcomes"?
- l. 058: "ablation as a necessary diagnostic tool for physics informed data generation", why would it be necessary?
- l. 156: what is "downstream learnability"?
- l. 202-205: "explicit energy-conservation enforcement has no measurable impact when the analytical formulation is physically consistent, with constrained and unconstrained generators achieving near-identical errors", hard to read / understand

---

### Meta-Review · Area_Chair_oEd4 · 2026-03-02

**Recommendation:** Reject

**Metareview:**

The paper does not seem a good fit to the workshop as it lacks deeper insights. The problem setup is interesting but the work in itself does not seem a good fit. I recommend a rejection.

---

### Decision · Program_Chairs · 2026-03-02

Reject